# Influence of Implant Material and Surface on Mode and Strength of Cell/Matrix Attachment of Human Adipose Derived Stromal Cell

**DOI:** 10.3390/ijms21114110

**Published:** 2020-06-09

**Authors:** Susanne Jung, Lauren Bohner, Marcel Hanisch, Johannes Kleinheinz, Sonja Sielker

**Affiliations:** Department of Cranio-Maxillofacial Surgery, Research Unit Vascular Biology of Oral Structures (VABOS), University Hospital Muenster, 48149 Münster, Germany; Susanne.Jung@ukmuenster.de (S.J.); lauren.bohner@ukmuenster.de (L.B.); Marcel.Hanisch@ukmuenster.de (M.H.); Johannes.Kleinheinz@ukmuenster.de (J.K.)

**Keywords:** hADSC, tissue regeneration, implants, matrix attachment, titanium, zirconia

## Abstract

A fundamental step for cell growth and differentiation is the cell adhesion. The purpose of this study was to determine the adhesion of different cell lineages, adipose derived stromal cells, osteoblasts, and gingival fibroblast to titanium and zirconia dental implants with different surface treatments. Primary cells were cultured on smooth/polished surfaces (titanium with a smooth surface texture (Ti-PT) and machined zirconia (ZrO_2_-M)) and on rough surfaces (titanium with a rough surface texture (Ti-SLA) and zirconia material (ZrO_2_-ZLA)). Alterations in cell morphology (f-actin staining and SEM) and in expression of the focal adhesion marker were analysed after 1, 7, and 14 days. Statistical analysis was performed by one-way ANOVA with a statistical significance at *p* = 0.05. Cell morphology and cytoskeleton were strongly affected by surface texture. Actin beta and vimentin expressions were higher on rough surfaces (*p* < 0.01). Vinculin and FAK expressions were significant (*p* < 0.05) and increased over time. Fibronectin and laminin expressions were significant (*p* < 0.01) and did not alter over time. Strength of cell/material binding is influenced by surface structure and not by material. Meanwhile, the kind of cell/material binding is regulated by cell type and implant material.

## 1. Introduction

Despite the high success rate of titanium and zirconia dental, a peri-implant inflammatory reaction can occur and lead to bone loss around implants. In these cases, regenerative procedures are usually required to restore the bone defect and to prevent the implant failure [1,2,3,4].

Although autogenous graft is still considered the gold standard for bone regeneration due to its capacity to promote osteogenesis, complexity and morbidity related to the surgical procedure are limiting factors for its use [5]. Therefore, the development of new bone substitutes by means of bioengineering techniques have been gaining special attention. Physiological components are able to induce bone formation by stimulating angiogenesis and cell differentiation. For instance, human adipose-derived stromal cells (hADSCs) can be differentiated in multiple lineages as adipogenic, chondrogenic, and osteogenic cells [6,7]. Our research group previously showed how titanium and zirconia dental implants stimulate the proliferation and osteogenic differentiation of hADSCs [8].

Nonetheless, tissue regeneration is a process involving not only hard but, also, soft tissues. Whereas hADSCs and primary human osteoblasts (hOB) are known for their bone differentiation properties, human gingival fibroblasts cells (HGFs) are essential for the regeneration of peri-implant soft tissue [9]. In this regard, a fundamental step for the cell growth and differentiation is its adhesion to a surface. Cell adhesion occurs due to a combination between chemical forces and gene expression, which, in turn, regulates cell migration [10]. Focal adhesion points are responsible not only by cells’ connectivity but, also, by the communication between intra and extracellular environments. Through the adhesion signalling system, the cytoskeleton is modulated, resulting in the reorganisation of the extracellular matrix, such as on the regulation of cell proliferation and differentiation [11].

Likewise, cell adhesion regulates the interaction between cells and implant surface, since the protein attachment to their receptor indicates how osteogenic cells interact with the dental implant surface [10]. However, how and in which orientation cells adhere to the material surface depends on their physicochemical characteristics. Implant surface structure can influence cell behaviour by the induction of mechanical signals, which can lead to a better organisation of the cytoskeleton and, subsequently, favouring the osteogenic differentiation of mesenchymal stem cells [9].

Considering that cell adhesion, such as related protein expressions, vary according to the material properties [12], the purpose of this study was to determine the adhesion of different cell lineages to titanium and zirconia dental implants presenting rough and smooth surfaces. Cell morphology and binding, such as the expression of the extra cellular matrix marker and focal adhesion markers, were analysed on the gene and on the protein level.

## 2. Results

### 2.1. Cell Morphology

Alterations on cells’ arrangement and morphology were summarised in Figure 1, Figure 2 and Figure 3. Pictures of day seven were presented, and no further morphological prediction was possible after day 14 because cells overgrew the surfaces. As shown in Figure 1 and Figure 2, hADSC and hOB morphology were significantly influenced by surface texture. Arrangement of the cytoskeleton differed on rough surfaces in comparison to the control group or machined/polished surfaces. On rough surfaces, the cell arrangement was random and nonoriented, and cell/cell or cell/matrix contacts via pseudopodia were distinct. Conversely, no significant difference was observed in the cell morphology of the control group and machined/polished surfaces. Likewise, the cytoskeleton arrangement was also similar. For HGFs, no significant differences between rough and machined/polished surfaces and the control group were visible (Figure 3). This was confirmed by the gene expression of ACTB and by the protein expression of a focal adhesion kinase, as reported in Figure 4. There was no alteration in expression over time and between tested surfaces or the control.

### 2.2. Cell Attachment

Changes in the gene and protein expressions of the focal adhesion marker and extracellular matrix marker were analysed by RT-qPCR and ELISA. Alterations in the fold change from day one to day seven and from day one to day 14 were shown in Figure 4. Statistical differences among groups were assessed by one-way ANOVA at *p* < 0.05.

Expression alterations were significant for actin beta (*p* = 0.006) and for vimentin (*p* < 0.01). As shown in Figure 4, at the fold change to day one, both markers were strongly expressed in hADSCs, being higher on rough surfaces in comparison to machined/polished surfaces. Furthermore, both markers showed the strongest fold change on rough titanium (Titan-SLA™). Conversely, the vimentin-typical fibroblast cytoskeleton element showed a strong expression for HGF and a weak expression for hOB. In HGF, a weak alteration in the expression of actin beta was observable (Figure 4), which corresponded with staining of the actin filaments (Figure 3).

Further, the expression alterations of vinculin, main structure protein of the focal contacts, and focal adhesion kinase (FAK), main marker of the focal contacts, were shown in Figure 4. Expression alterations of vinculin (*p* < 0.01) and FAK (*p* = 0.042) were statistically significant. In hADSC, both markers showed a strong expression that increased over time. The expression of vinculin was higher on titanium, whereas the expression of focal adhesion kinase was higher on zirconia (Figure 4). The numbers of focal contacts and, consequently, strength of cell/material binding increased in hADSC over time. In hOB, the expression of vinculin did not significantly alter over time. Controversially, the expression of FAK increased strongly on titanium with rough surfaces (Figure 4). In HGF, no alterations in the expression of vinculin and FAK were observed (Figure 4). This corresponds with the unchanged cell morphology and cytoskeleton of HGF on rough or machined/polished surface (Figure 3).

As members of the extracellular matrix, the expression alterations of fibronectin and laminin were analysed (Figure 4). Both are glycoproteins and involved in cell adhesion. The expression alterations of fibronectin (*p* = 0.0002) and of laminin (*p* < 0.01) were significant. For hOB and HGF cells, fibronectin expression was not dependent on time, material, or surface texture. Likewise, for hADSCs and HGF cells, the laminin expression did not alter over time, such as in relation to the material or surface. Titanium surfaces showed the strongest alterations (Figure 4).

## 3. Discussion

In this study, a systematic comparison of mesenchymal stem cells and edaphic cells on dental implant materials and surface textures was performed. The behaviour of hADSCs, a promising cell source for guided oral tissue regeneration, and edaphic cells, as hOB and HGF, on titanium and zirconium dioxide with smooth and rough surface textures were systematically analysed. Previously, we demonstrated that the cell viability and proliferation rate were neither disturbed by titanium nor zirconium dioxide [8]. Observed cytotoxic effects were insignificant. In summary, the cell vitality and proliferation rate were higher on zirconium dental implants, especially when presented on rough surfaces. It is assumed that this texture showed better growth conditions as a polished surface. In addition, the cell vitality and proliferation rate were higher in hADSCs when compared to hOB and HFG. Previously, a loss in the stem cell potential and a beginning transdifferentiation in osteoblast-like cells were observed in hADSCs. This was induced by implant material and surface texture alone [8]. In the present study, we looked closely at effects related to cell morphology and effects related to the kind and strength of cell attachment to the implant material and surface texture.

The cell morphology of hADSC and hOB were affected by surface structure. On rough surfaces, the cell morphology and cytoskeleton arrangements were altered compared to the machined/polished surface. There were slight alterations between the control group and the machined/polished surfaces. A prediction about used material was not possible. The surface structure seemed to be the factor of influence. The alteration of the cell morphology by the surface texture has been observed before. Lohmann et al. characterised osteoblast-like cells on titanium with a rough surface texture (Ti-SLA™) as more rounded cells, with filopodia extending into the micropits on the surface. Unlike cells on titanium with a smooth surface texture (Ti-PT), as flattened cells and the cell morphology were comparable with cells on cell culture plate surfaces [13]. This was also observed, i.e., for osteosarcoma cell line MG-63 and osteoblast precursor cells MC3T3-E1 [12,14]. Kim et al. pictured an increased expression of beta actin in osteoblasts grown on titanium with rough surfaces [15]. Similar observations were made for bone marrow stem cells. The cell morphology was more spread on smooth surfaces [16], and the cell morphology of stem cells isolated from dental pulps was on rough surfaces more random and on smooth surfaces more aligned [17]. 

Interestingly, the cell morphology of HGF cells was not influenced by surface structure or material, since this was similar on all tested surfaces. The nonalteration of HGF cell morphology by surface texture was also described in other studies [18,19,20]. In contrast, Größner-Schreiber et al. described altered cell morphology by the grade of titanium surface roughness and a more aligned morphology on smooth surfaces [21]. 

A fundamental step in tissue regeneration is cells binding to the implant surface. In order to optimise the regeneration, it is important to understand the cell binding strength to the implant material surface. When thinking about guided tissue regeneration, hADSC is a promising cell source. However, it is still unclear how hADSC binds to a dental implant surface, such as if the binding is comparable with hard tissue cells, as osteoblasts, or with soft tissue cells, as gingivafibroblasts. This study put us in a position to compare directly hard tissue cells (hOB), soft tissue cells (HGF), and alternative mesenchymal stem cells (hADSC). Cell adhesion is regulated by the extracellular matrix (ECM). Cells adhere to the ECM via an integrin-mediated adhesion that links the EMC to the cytoskeleton, and integrins serve as receptors for ECM proteins [22].

In this study, the main and typical markers of ECM (fibronectin, laminin, focal adhesion kinase, and vinculin) and cytoskeleton markers (actin and vimentin) were analysed. Furthermore, the expression alterations between the cell types and implant materials or surface structures were compared. One result of this study is that the kind of cell binding to the material via focal contacts is heterogeneous. Like the observed effects of cell morphology, it seems that the surface structure is the factor of influence and not the implant material. Generally, marker expressions increased stronger on rough surfaces compared to smooth surfaces. In the control group, the expression of markers decreased over time. These results endorse the conclusion that the strength of the cell/material binding increases on rough surfaces compared to smooth surfaces. Further studies with osteoblasts confirmed this prediction, showing a relation between the surface roughness and cell adhesion [14,23]. 

One marker for strengthening the cell adhesion to the dental material surface is an increasing FAK expression, which was shown in osteoblasts growing on rough titanium surface textures [24]. A continuous increase in FAK and vinculin expressions was observed before. Furthermore, the significantly higher FAK expression on rough surfaces compared to the smooth one was previously described [24,25,26]. In our study, we observed an increased expression rate of FAK in hOB on rough titanium (Ti-SLA™). The expression of FAK in HGF cells remained constant. This correlates with the nonalteration of the cell morphology by rough surface textures. This was confirmed by another study in which FAK expression in HGF cells growing on titanium surfaces with different textures was steady [27]. However, the strongest increased expression of FAK was observed in hADSC. FAK plays an important role in promoting osteoblast focal adhesion formations on implants [28]. It seems that the expression of FAK is upregulated during osteo-specific differentiations, and it regulates the synthesis of osteo-specific proteins [24,26]. This correlates with hints for a beginning osteogenic differentiation of hADSC cells and an increased expression of osteo-specific proteins [8]. 

Apart from this, the kind of cell/material binding seems to be regulated by the implant material. The cell/material binding differs between hard and soft tissue cells. In this comparative study, we observed differences in the expression patterns between hOB and HGF cells. Whereas many studies investigated the osteoblast binding to the implant material, there is no sufficient evidence about the kind of cell/material binding of hADSC to either dental implant materials or surface structures. According to the authors’ knowledge, this is the first study assessing the hADSCs’ binding. In osteoblasts, the cell/material binding occur in 60–70% via fibronectin and in 40–50% via laminin [29]. We observed an unchanged expression of fibronectin and an increased expression of laminin with the strongest alteration rate on titanium. Gronthos et al. observed similar alterations in expressions. The expression of laminin increased, while the expression of fibronectin laminin remained constant. They postulated binding via laminin as a second type of binding and as an enhancement of binding via fibronectin [30].

In hADSC, laminin expression remains constant, whereas fibronectin expression increases over time and with the strongest alteration on zirconia material (ZrO2-ZLA™). In HGF cells, both laminin and fibronectin expressions remained constant, which was already confirmed in other studies. Only slight differences in the gene expression of fibronectin was found comparing HGF cells growing on Laser-Lok, titanium, and zirconia surfaces [31]. Further, a significant downregulation of vimentin and fibronectin was found between polished titanium and rough modified titanium/zirconia surfaces [32]. Contrary results were described by Miao et al. (2017). They observed an increased gene expression of laminin, fibronectin, and vinculin in HGF cells growing on titanium with a rough surface texture compared to a smooth surface structure [33]. All three analysed cell types behaved in a varying manner, so that a favoured prediction about hADSC behaviour in comparison to edaphic cells, as hOB and HGF, was not possible. 

## 4. Materials and Methods

### 4.1. Study Design and Ethical Approval

The study evaluated cell morphology and cell binding of hADSCs, hOBs, and HGFs on titanium and zirconia dental implants with polished/machined and rough surfaces. Cell morphology was recorded based on staining of the cytoskeleton and SEM analysis, whereas cell binding was recorded based on gene and protein expression analysis. Experimental design was conducted in accordance with the “Declaration of Helsinki” and approved by the Ethics Committee of the Faculty of Medicine, University of Muenster (#2016-624-f-S). Previous to the cell isolation, a written informed consent was obtained from all donors. 

### 4.2. Main Cell Culture

Isolation and culture techniques of human primary cell cultures were described before [8,34]. Additionally, as the main cell culture [8]. Cells were cultured on titanium and zirconia discs (Straumann, Basel, Switzerland) measuring 5 mm in diameter, according to the following groups: (1) titanium implants with a polished surface (Ti-PT), (2) sandblasted and acid-etched titanium (Ti-SLA™), (3) sandblasted and alkaline-etched zirconia (ZrO2-ZLA™), and (4) machined zirconia (ZrO2-M).

For control, cells were cultivated as a monolayer in cell culture plates. Samples were analysed one, seven, and 14 days after the beginning of the experiment. Cell culture part was repeated three times.

### 4.3. Morphology Assay 

For the f-actin staining of the cytoskeleton cytopainter, Phalloidin iFluor 488 reagent (Abcam, Cambridge, UK) was used. Cells were fixed with 4% formalin in phosphate-buffered saline (PBS) without methanol according to the manufacturer’s protocols. Staining was performed directly with a 1:1000 dilution in 1% bovine serum albumin (BSA) for osteoblasts, a 1:700 dilution for hADSC, and a 1:500 dilution for HGF. Stained cells were examined with the fluorescence microscope Axioplan 2 (Carl Zeiss, Jena, Germany). For SEM analysis, cells cultures were fixed in glutaraldehyde (4% in phosphate buffer, pH 7.4); washed with PBS; and dehydrated in a graded ethanol series (30%, 50%, 70%, 90%, 96%, and absolute). Then, the samples were subjected to critical-point drying with liquid CO_2_ according to the standard procedure. Subsequently, the samples were mounted on an aluminum specimen holder by using conductive adhesive tabs (Plano, Wetzlar, Germany) and sputter-coated with a gold layer having a thickness of approximately 15 nm. Imaging was performed with a S800 SEM (Hitachi Ltd., Tokyo, Japan) at the Institute of Medical Physics and Biophysics (Division: Electron Microscopy, University of Muenster).

### 4.4. Gene Expression and Protein Expression Analysis

Method for gene expression analysis and for protein expression analysis was descripted before [8]. The used primers for RT-qPCR and ELISAs are listed in Table 1. Expressions factors were statistically analysed by one-way ANOVA (*p* < 0.05). Post-hoc analysis was performed with Bonferroni-Holm test (Daniel’s XL Toolbox version 6.53; http://xltoolbox.sourceforge.net) [35].

## 5. Conclusions

Cell morphology is affected by surface texture. A distinct statement on differences in mode and strength of cell/matrix attachment between titanium and zirconium dioxide could not been made. However, kind of attachment is regulated by cell type and used material. And strength of attachment is affected by surface texture.

## Figures and Tables

**Figure 1 ijms-21-04110-f001:**
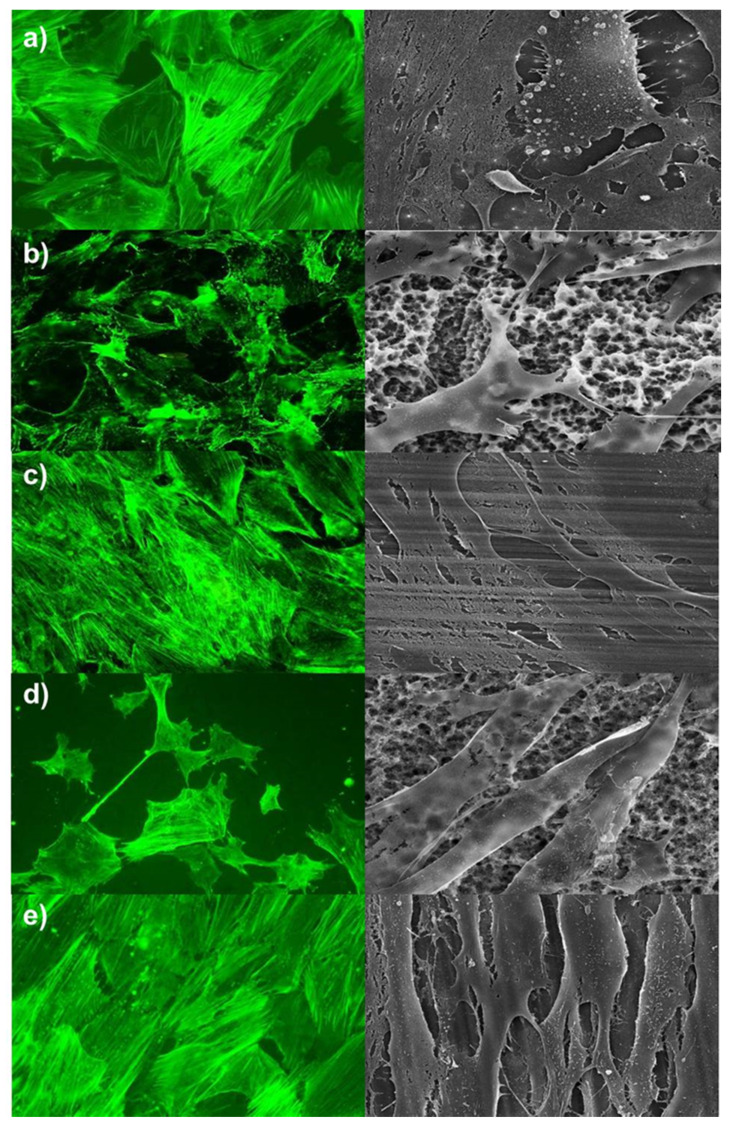
Alterations in the cell morphology of human adipose-derived stromal cells (hADSCs). Left site = f-actin staining, magnification factor 100×; right site = SEM, magnification factor 1200×. (**a**) Control, (**b**) rough titanium (Titan-SLA), (**c**) titanium with a smooth surface texture (Titan-PT), (**d**) zirconia material (ZrO_2_-ZLA), and (**e**) machined zirconia (ZrO_2_-M).

**Figure 2 ijms-21-04110-f002:**
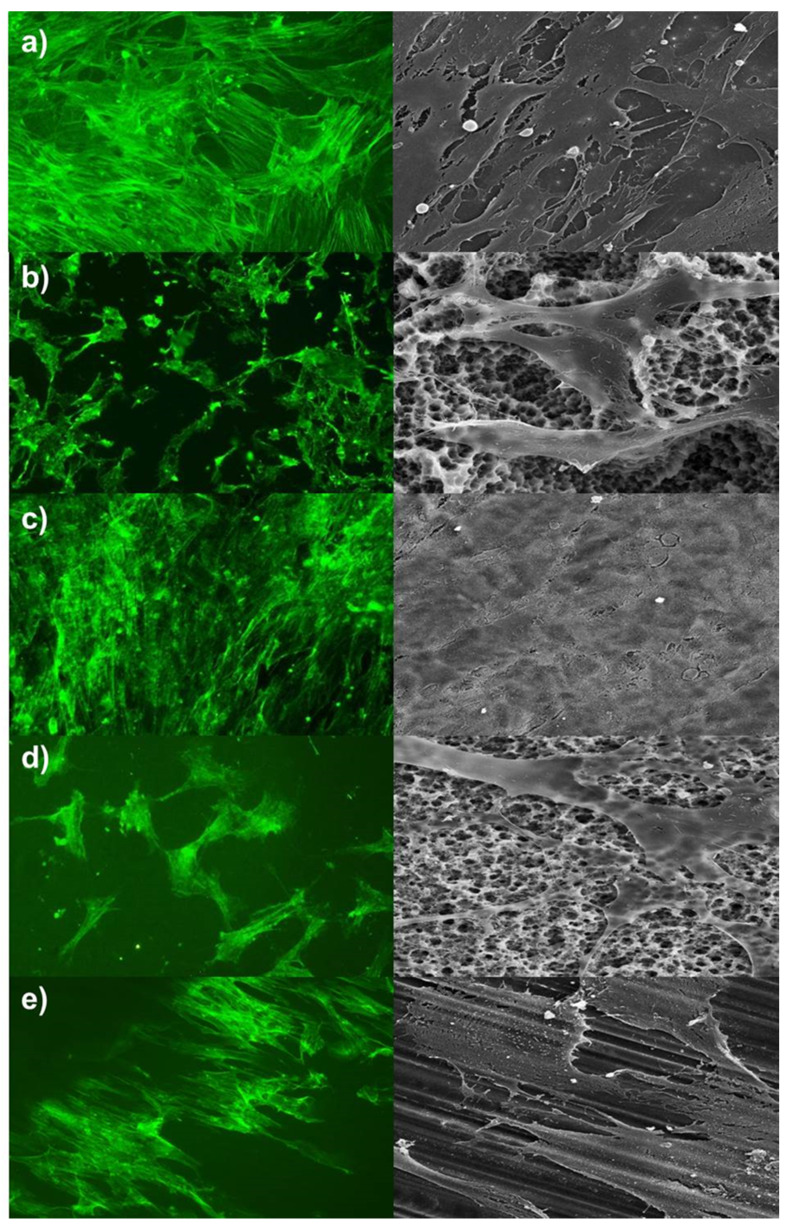
Alterations in the cell morphology of human osteoblasts (hOB). Left site = f-actin staining, magnification factor 100×; right site = SEM, magnification factor 1200×. (**a**) Control, (**b**) Titan-SLA, (**c**) Titan-PT, (**d**) ZrO_2_-ZLA, and (**e**) ZrO_2_-M.

**Figure 3 ijms-21-04110-f003:**
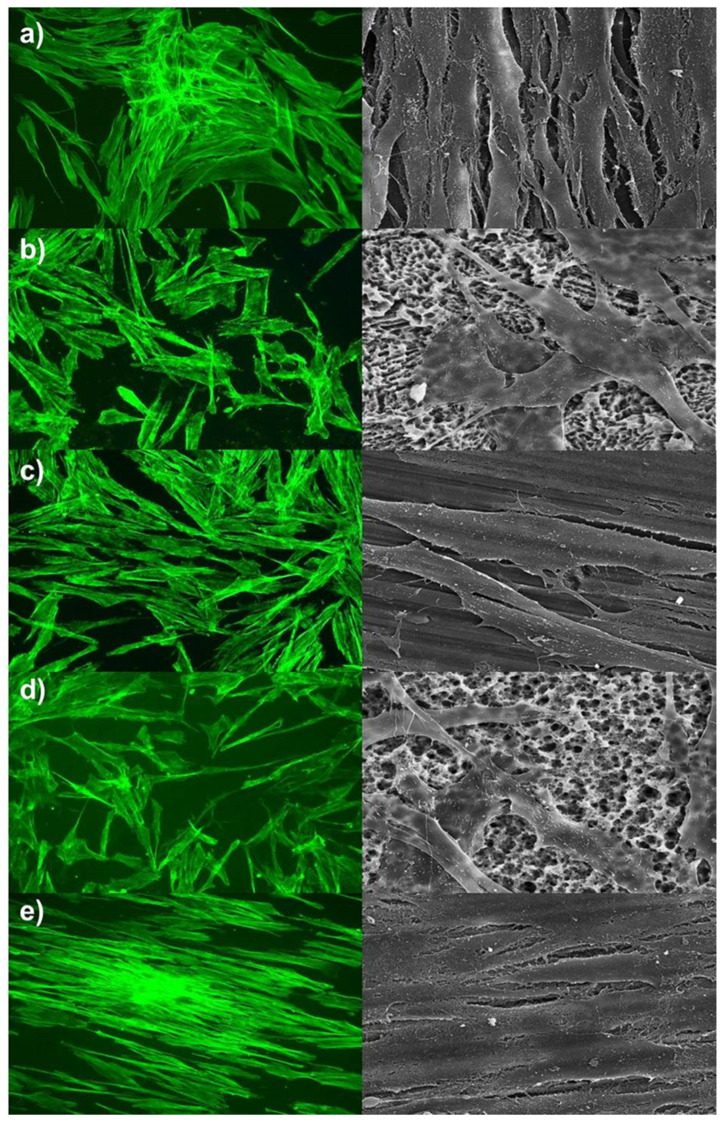
Alterations in the cell morphology of human gingival fibroblasts cells (HGFs). Left site = f-actin staining, magnification factor 100×; right site = SEM, magnification factor 1200×. (**a**) Control, (**b**) Titan-SLA, (**c**) Titan-PT, (**d**) ZrO_2_-ZLA, (**e**) ZrO_2_-M.

**Figure 4 ijms-21-04110-f004:**
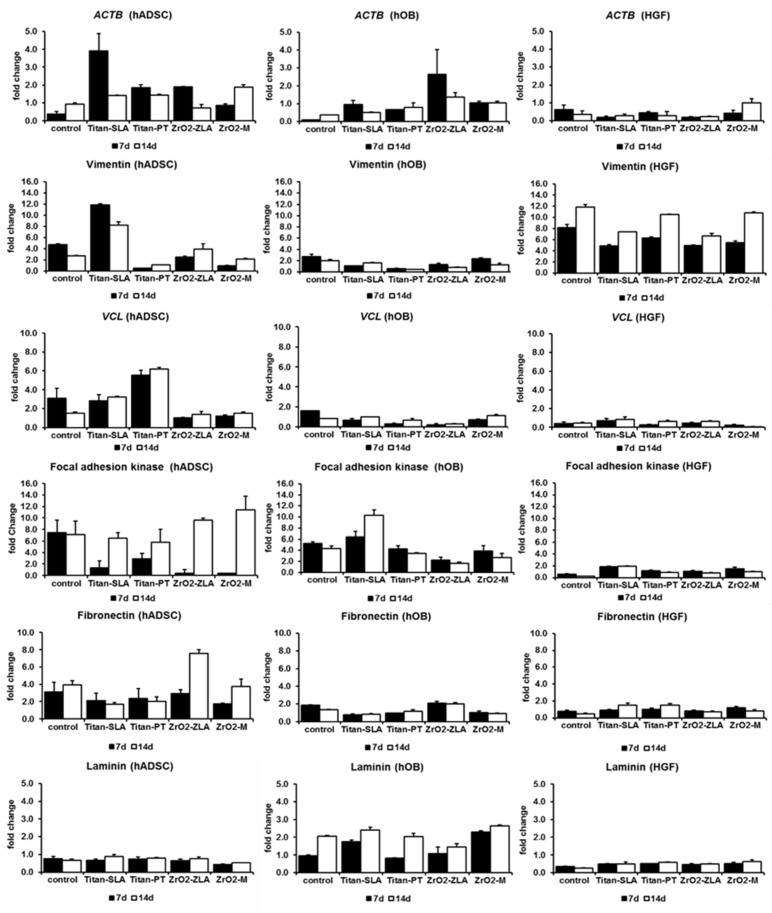
Fold changes of the focal adhesion and extra cellular matrix markers compared to day 1 (standard abbreviation as error).

**Table 1 ijms-21-04110-t001:** Primers used for RT-qPCR (Qiagen, Hilden, Germany) and ELISA kits (Abcam, Cambridge, UK).

Gene/Protein	Primer	ELISA
Focal Adhesion
VCL	PPH02077F	
ACTB	PPH00073G	
Vimentin		ab173190
Focal Adhesion Kinase		ab187395
Extracellular Matrix
Fibronectin		ab219046
Laminin		ab119599
Housekeeping Genes
RPLP0	PPH21138F	
GAPDH	PPH00150F

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
