# Peer review of "Influence of Implant Material and Surface on Mode and Strength of Cell/Matrix Attachment of Human Adipose Derived Stromal Cell"

_ijms, 2020, doi:10.3390/ijms21114110_

Round 1
Reviewer 1 Report
Dear authors,
The comparison results of mesenchymal stem cells and edaphic cells on the rough and smooth implant surfaces, in this manuscript, in conjunction with the results for cell viability and proliferation rate in your previously published paper, Ref. 8, is very fascinating. To improve the current version of your paper, please apply the following comments accordingly:
1- Both abstract and introduction sections should be rewritten. The literature study on the computational modeling and simulation on this topic is absent from the current version (Take a look: ACS applied materials & interfaces, 11(45), 41906-41924). What I expect here is a detailed analysis of the current trend in computational studies and a comparison between your results (if possible) and numerical studies.
2- Section 4 (line 205 to 282) is copied from your previously published paper (Ref. 8) with some modifications. Please summarize this section and transfer it to the supplementary material.
3- Similar to comment 1, the discussion section should be improved and the computational results must be fully analyzed.
Author Response
Comment #1 & comment #3:
Comparative study to Frahs et al., 2019 (Prechondrogenic ATDC5 Cell Attachment and Differentiation on Graphene Foam; Modulation by Surface Functionalization with Fibronectin; ACS Appl. Mater. Interfaces 2019, 11, 41906−41924; DOI:10.1021/acsami.9b14670):
Thank you very much for this interesting publication. It is well written and presenting new findings. If I understand rightly, whole manuscript should be reanalysed to and like giving publication.
I have to deny. At the first look both studies giving attention to cell behaviour to implant materials. But with a second closer look, both studies are completely different. Giving publication analysed a new scaffold type. They analysed effects of graphene foam modified with fibronectin to enhance cell proliferation and cell attachment. In our study, we analysed in this study part, cell attachment to common used dental material with just topographic surface modification. In discussion, we focused on related studies with comparable study designs. Further whole study design was designated, to analyse effects to common implant materials and “simple” topographic surface modification. We excluded knowingly studies with (bio-) chemical modified surfaces. With this intention, we tried to stay focused and get not lost in quantify of good and interesting studies. Further, we compared human adipose derive stromal cells with oral primary cells, osteoblasts and gingiva fibroblasts.
Reviewer indicates that introduction and discussion should be more detailed and results should be compared stronger with literature. Also, I have to deny. In respect to our study design and analysed cell types, we cited good and significant references in introduction and in discussion. Most studies with comparable study design analysed cell behaviour and cell proliferation on dental implant materials. Only few analysed cell attachment and when they did, they analysed in first line osteoblast followed by gingiva fibroblast and fewer studies analysed adipose derived stromal cells.
#2 Materials and Method:
Thank you very much for your comment. You are absolutely right. Most of the information could be found in our first study part. We summarized part 4 “Materials and Methods” and added only essential information, which are mandatory for readership.
Subitem 4.2 “Isolation of primary human cell culture” and 4.7 “Statistical analysis”, and Table 1 was deleted. Further, subitems were modified and overlapping parts were deleted. Citation to reference #8 was added.
Reviewer 2 Report
Very well designed, performed, and described research.
I dont't have any comments. Good job!
Author Response
Thank you very much. We always work hard to optimize our work. It is good to hear, that we are on the right way.